# Diet Quality and Resilience through Adulthood: A Cross-Sectional Analysis of the WELL for Life Study

**DOI:** 10.3390/nu16111724

**Published:** 2024-05-31

**Authors:** Sparkle Springfield-Trice, Cara Joyce, Yi-Hsuan Wu, Ann W. Hsing, Kristen Cunanan, Christopher Gardner

**Affiliations:** 1Department of Public Health Sciences, Parkinson School of Public Health Sciences and Public Health, Loyola University Chicago, 2160 S 1st Ave, Maywood, IL 60153, USA; 2Department of Medicine, Stritch School of Medicine, Loyola University Chicago, 2160 S 1st Ave, Maywood, IL 60153, USA; cjoyce6@luc.edu; 3Stanford Prevention Research Center School of Medicine, Stanford University, 3180 Porter Drive, Palo Alto, CA 94304, USA; joyceywu@stanford.edu (Y.-H.W.); annhsing@stanford.edu (A.W.H.); cgardner@stanford.edu (C.G.); 4Quantitative Sciences Unit, School of Medicine, Stanford University, 3180 Porter Drive, Palo Alto, CA 94304, USA; kcunanan@stanford.edu

**Keywords:** resilience, diet quality, adults, cross-sectional, WELL

## Abstract

Despite evidence suggesting the importance of psychological resilience for successful aging, little is known about the relationship between diet quality and resilience at different ages. Our study aims to examine the association between diet quality and resilience across the stages of adulthood. Using Stanfords’ WELL for Life (WELL) survey data, we conducted a cross-sectional study of diet quality, resilience, sociodemographic, perceived stress, lifestyle, and mental health factors among 6171 Bay Area adults. Diet quality was measured by the WELL Diet Score, which ranges from 0–120. A higher score indicates a better diet quality. Linear regression analysis was used to evaluate the association between the WELL Diet Score and overall resilience and within the following age groups: early young (18–24), late young (25–34), middle (35–49), and late adulthood (≥50). To test whether these associations varied by age groups, an age group by resilience interaction term was also examined. In the fully adjusted model, the WELL Diet Score was positively and significantly associated with overall resilience (all ages (β = 1.2 ± sd: 0.2, *p* < 0.001)) and within each age group (early young (β = 1.1 ± sd: 0.3, *p* < 0.001); late young (β = 1.2 ± sd: 0.3, *p* < 0.001); middle (β = 0.9 ± sd: 0.3, *p* < 0.001); and late adulthood (β = 1.0 ± sd: 0.3, *p* < 0.001)). Young adults demonstrated the strongest associations between diet quality and resilience. However, there were no significant age-by-resilience interactions. Diet quality may be positively associated with resilience at all stages of adulthood. Further research is needed to determine whether assessing and addressing resilience could inform the development of more effective dietary interventions, particularly in young adults.

## 1. Introduction

Diet-related diseases, such as cardiovascular disease (CVD), are among the leading causes of death in the United States [1]. Despite the high prevalence of CVD, most Americans do not adhere to evidence-based dietary guidelines for chronic disease prevention [1,2]. As such, poor diet quality is the most prevalent CVD risk factor in US adults [3,4,5,6,7,8,9,10]. Although age is a well-established risk factor for CVD, it is less commonly recognized that diet quality tends to improve as we progress through subsequent phases of life (e.g., late young adulthood, 25–34 years; middle adulthood, 35–49 years; and late adulthood, 50+ years) [11,12,13]. Thus, enhancing diet quality in early adults is important for preventing and managing CVD risk factors throughout the life course [14,15,16]. 

High stress adversely affects dietary behaviors, promoting appetite and a preference for foods that are high in fat and sugar. Also, multilevel barriers have made it stressful for Americans to engage in healthy eating [17,18,19]. Over the past few decades, the food system has been flooded with aggressively marketed ultra-processed foods and sugar-sweetened drinks, making it psychologically taxing to engage in healthy eating [20,21,22,23,24]. Thus, identifying psychological assets that can serve as a stress buffer could be a key protective strategy to improving and sustaining a better diet quality, particularly in vulnerable populations [25,26].

Building psychological resilience—the perceived ability to bounce back from stress—may lead to better diet quality. Evidence suggests that psychological resilience (hereafter resilience) increases with age [27,28] and is positively linked to better diet quality in both young and older adults [29,30,31,32,33,34,35], although recent studies have focused on international populations. Compared with a younger person, it stands to reason that an older person may have accumulated more resources and developed greater abilities to cope with stress [36,37,38]. Moreover, behavioral interventions, including resilience training (e.g., meditation, mindful breathing, gratitude, and forgiveness exercises) have been shown to improve diet and diet-related outcomes in older adults [39,40,41,42]. However, definitions and measures of resilience vary in these studies, and evidence in young adults is limited, with even less known about the relationship between diet quality and resilience across adulthood.

To address this gap in the literature, our study aims to examine the relationship between diet quality and resilience while adjusting for perceived stress, sociodemographic, lifestyle, and mental health factors within the young, middle, and late adults enrolled in the Stanford WELL for Life (WELL) study. We hypothesize that resilience will show significant and positive associations with diet quality across age groups, as measured by the WELL Diet Score. Our study explored the link between age, resilience, and diet quality—an understanding that may inform the development of more effective dietary interventions to prevent disease (e.g., CVD) and promote health at varying stages of adulthood.

## 2. Methods

### 2.1. Study Design

Our study is a cross-sectional secondary analysis of Stanford WELL survey measures that assesses resilience, perceived stress, and diet-related behaviors. Since it was created within the Stanford Prevention Research Center (SPRC) in 2017, investigators have been using WELL to generate comprehensive scientific data to help define, understand, and improve well-being among people from diverse backgrounds [43]. The WELL study design, protocol, informed consent measures, recruitment, and survey measures have been published elsewhere [43,44,45,46]. Briefly, the WELL Survey consists of 76 items, spanning 10 domains of well-being. These domains include measures for stress and resilience, experiences of emotions (positive and negative feelings), lifestyle behaviors (physical activity and diet-related behaviors (used to calculate the WELL Diet Score)) as well as demographic and medical history (including self-reported height and weight, hypertension, depression). As of June 2021, 6171 women and men (18 years of age or older) have completed the survey. Participants provided informed consent and the study was approved by the Stanford University Institutional Review Board.

### 2.2. Measures

#### 2.2.1. Main Outcome

Diet quality is measured by the WELL Diet Score. A detailed description of WELL Diet Score development was previously published [46]. Briefly, following the 2015 evidence-based dietary guidelines for chronic disease prevention, SPRC’s public health professionals developed the WELL Diet Survey (12 items). The WELL Diet Score was assessed by eliciting information regarding dietary intake and meal preparation behaviors.

The WELL Diet Score, ranging from 0–120, is the sum of 12 diet-related items that are rated from 1–10. Scoring criteria is based on the frequency of consumption, with 10 representing the maximum score. Such that, a higher total score indicates a better diet quality (and adherence to evidence-based guidelines). WELL Diet Score items included (1) vegetables, (2) fruits, (3) whole grains, (4) beans or lentils, (5) sugar-sweetened beverages (including 100% fruit juice) (reverse-scored), (6) red/processed meats (reverse-scored), (7) nuts and seeds, (8) high-sodium processed foods (reverse-scored), (9) sugar-sweetened baked goods or candy (reverse-scored), (10) fish, (11) preparing meals at home, and (12) eating fast food (reverse-scored).

#### 2.2.2. Primary Exposure: Resilience

An adapted version of the Brief Resilience Scale and the Connor–Davidson Resilience Scale were used to assess resilience (e.g., “How confident are you that you can bounce back quickly after hard times?” and “How confident are you that you can adapt to change?”). Respondents were asked to indicate their level of confidence on nine items, ranging from “extremely confident” to “not at all confident”. The Resilience Score is the mean value of the nine items and was standardized to range from 0–10. See Appendix A for a detailed description of the resilience items. 

#### 2.2.3. Covariates: Sociodemographics, Perceived Stress, Lifestyle (and BMI), and Mental Health

Sociodemographic Variables: Age (years), gender, race/ethnicity (White/Caucasian, Hispanic, Black/African American, Asian, and Other), education (HS or below, some college, college graduate, or graduate), and marital status (married, living with a partner, single, or other) were collected via self-reporting.

Perceived Stress: This was assessed through four items (an adapted version of the Global Measure of Perceived Stress). For example: “During the last two weeks, how often have you felt that you were not able to give enough time to the important things in your life, or handle the problems you are experiencing?” Respondents were then asked to indicate the frequency of their feelings for each item, ranging from “never” to “very often.” Scores were standardized to range from 0–10. This was reverse-scored such that higher scores indicate lower levels of perceived stress.

Lifestyle (physical activity, smoking status) and BMI variables: Physical activity was assessed through an adapted version of the International Physical Activity Questionnaire and the Stanford Leisure-Time Activity Categorical Item. Participants were asked to consider their physical activity by answering the following question: “During the past month, which statement best describes the kinds of physical activity you usually did? Do not include the time you spent working at a job”. They were given six statements to choose from, ranging from “I did not do much physical activity. I mostly did things like watching television, reading, playing cards, or playing computer games. Only occasionally, no more than once or twice a month, did I do anything more active such as going for a walk or playing tennis” to “Almost daily, that is, five or more times a week, I did vigorous activities such as running or riding hard on a bike for 30 min or more each time”. The scores ranged from 0–10. Smoking status was assessed through the options of “never”, “former”, or “current”; hypertension was self-reported (y/n); and BMI was calculated from self-reported weight and height. 

Mental health (depression, positive and negative affect, and wellbeing): History of clinical depression was also self-reported, namely: “Have you ever been told by a doctor or other health professional that you had depression?” (y/n). Positive affect was assessed by a six-item inventory (designed by the WELL team), e.g., “During the last two weeks, how often did you feel content?” Respondents were then asked to indicate the frequency of their feelings for each item, ranging from “never” to “very often”. Negative affect was gauged by a five-item inventory (also WELL designed and reverse-coded). For instance, “During the last two weeks, how often did you feel sad?” Participants were then asked to indicate the frequency of their feelings for each item, ranging from “never” to “very often”. For both positive and negative affect, scores were standardized to range from 0–10. The total WELL Score was tabulated as a measure of well-being by combining scores for all 10 WELL domains, including social connectedness, lifestyle behaviors, physical health, stress and resilience, experience of emotions, purpose and meaning in life, sense of self, financial security, spirituality and religiosity, and exploration and creativity. Scores ranged from 0–100. 

### 2.3. Statistical Analysis

Age groups were defined as: 18–24 (early young adulthood), 25–34 (late young adulthood), 35–49 (middle adulthood), and ≥50 (late adulthood). Descriptive statistics were used to describe participant characteristics overall and by age group. Pearson’s correlation coefficients were calculated for bivariate associations between age (in years), the WELL Diet Score, and the Resilience Score. Linear models regressed the WELL Diet Score on age, Resilience Score, and an age-group-by-resilience interaction term. Overall age-group-stratified linear models were presented following progressive covariate adjustments, including (1) sociodemographics (gender, race, and education), (2) perceived stress, (3) lifestyle and BMI (smoking, physical activity, and BMI), and (4) mental health (history of depression, and positive affect). Note that the total WELL Score was not included in the statistical models. Collinearity was monitored using variance inflation factors. Analyses were performed using SAS version 9.4 (SAS Institute, Cary, NC, USA). 

## 3. Results

Our study included 6171 adults with a mean age of 39 years (38.8; see Table 1). On average, WELL participants were predominantly white (60.9%), female (71.7%), college-educated (69.1%), had a healthy BMI (18.5–24.9) (54.7%), and were non-smokers (96.4%). Nearly half the participants were married or living with a partner, with about 4% of the early young and 65% of the late adult quartiles being married. Twenty-four percent of our participants reported being depressed; similar percentages were found in each stage of adulthood. Approximately 13% of participants reported hypertension, 4% of which were early young adults and 30% of which were late adults. Overall, WELL participants reported moderate levels of perceived stress (reverse-scored) (mean of 5.5 out of 10 for the early young adults and mean of 6.5 out of 10 for late adults). Similarly, WELL participants in all stages of adulthood had moderately high perceptions of their resilience (mean of 6.7 out of 10 for the early young adults and mean of 7.0 out of 10 for late adults). Both positive and negative emotion (reverse-scored) scales demonstrated similar trends to resilience. WELL Diet Scores ranged from 58.7 in the youngest adults to 80.2 in late adulthood, with a mean for all ages of about 70 out of 120, suggesting that the overall diet quality could be improved in all age groups. 

Young adults (both early and late) reported WELL Diet Scores above and below the median of the total sample (which was 71 (IQR: 56–85)) compared with middle and late adults.

We found that both resilience and WELL Diet Scores were higher in older age groups, and the latter was significantly and moderately correlated with age (r = 0.40; *p* < 0.001). In terms of resilience, there was a significant positive and weak correlation with age (r = 0.11; *p* < 0.001). WELL Diet Scores and resilience also had a significant and positive correlation (r = 0.25; *p* < 0.001). Perceived stress had negative and significant correlations with age (r = 0.23), the WELL Diet Score (r = 0.29), and resilience (r = 0.53), respectively (see Appendix A).

As indicated in Table 2, all models demonstrated a significant and positive relationship between diet quality and resilience. The crude assessments have beta-coefficients of 2 across all age groups, and they attenuate with each successive addition to the model such that the beta-coefficients become smaller until the values for all age groups are close to 1. In the fully adjusted model, resilience remained significantly and positively associated with diet quality among all ages, after adjustment for perceived stress, sociodemographic, lifestyle and BMI, and mental health variables. Note all models were statistically significant, with the fully adjusted models showing the strongest association between diet quality and resilience in young adults. There were no significant interactions with age group (see Figure 1). 

## 4. Discussion

Our cross-sectional study examined the associations between diet quality and resilience among participants enrolled in the WELL study (early young, late young, middle, and late adulthood). These participants were predominantly white, college-educated, and relatively healthy females from California’s Northern Bay Area. We found significant associations between diet quality and resilience overall and within each age group. Although the associations were stronger among young (18–24 and 25–34 years of age) compared with middle-aged and older adults (35–49 and 50+ years of age), in the fully adjusted model of our sample, we did not observe a significant interaction between age and resilience. Our findings add to the theoretical and empirical evidence suggesting that perceived resilience, perceived stress, and diet quality increase with age and are therefore modifiable. Considering that addressing mental health in behavioral intervention research is becoming increasingly important, resilience may be an important starting point in the development of effective dietary interventions for CVD prevention at all ages, but especially among early young adults (18–24) [47,48,49].

Over the past decade, several cross-sectional studies have found a significant and positive association between diet quality and resilience [29,30,31,32,33,34,35]. These findings extended to multiple subpopulations, including both older and younger adults in the US and abroad. For example, using data from a large sample of predominantly white older women (mean age 77 years N = 77,395) enrolled in the Women’s Health Initiative, Springfield and colleagues (2020) found that high levels of resilience (as measured by the Brief Resilience Scale) were associated with CVD-related protective factors, including 22% greater odds of having a better diet quality (as measured by the Healthy Eating Index 2015 (HEI-2015)) (OR = 1.22 (95% confidence interval (1.15–1.30)) after adjusting for perceived stress, race/ethnicity, level of education, BMI, hypertension, diabetes, and high cholesterol [29]. Likewise, Lutz and colleagues (2017) found that young male and female adult Army and Air Force recruits (mean age 21 years; N = 656) with high resilience (as measured by the Connor–Davidson Scale) reported a higher diet quality (measured by the Healthy Eating Index 2010 (HEI-2010)) after adjusting for age, sex, race and ethnicity, education, smoking, and BMI [35]. 

Internationally, a large cross-sectional analysis of 10,812 Italian middle-aged adults recruited in the Moli-sani study (mean age 35 yrs., 52% female) showed that diet quality (as measured by the Greek Mediterranean Diet Index, Italian Mediterranean Diet Index, and olive oil and vegetables pattern) was positively associated with resilience (per the Connor–Davidson Scale) [30]. Roberts and colleagues (2022) reported similar findings in a French sample of predominantly older white women enrolled in the NutriNet-Santé Study (73.5% female, mean age 55.4 yrs.). They found that resilience (per the Brief Resilience Scale) was significantly and positively associated with diet quality (as measured by the modified French National Nutrition and Health Program Guideline Score and the NOVA classification system for ultra-processed foods [31,50].

Despite decades of research conceptualizing the broader relationships between health behavior, stressors, coping, and resilience, little is known about the potential mechanisms relating resilience to diet quality [51]. Individuals who can effectively cope with stressors (e.g., emotional distress and low access to high-quality foods) are more likely to practice healthier dietary behaviors than those who cannot [52]. In contrast, healthier diets may have a more positive effect on brain processes and mental health through the gut microbiome [53,54,55]. More research is needed to understand these biopsychological pathways, as well as how they can build on existing resilience-related positive psychology constructs to improve diet quality, particularly in the context of psychological distress [56].

Despite our sample’s social advantages (e.g., high education, physical activity, non-smoking, and relative well-being), they reported high levels of stress and depression (pre-pandemic). Approximately 25% of our participants reported a history of depression, and these percentages may be significantly higher than national averages [57,58]. Based on self-reported data from 2021 SAMHSA reports, women had a higher prevalence of major depressive episodes (10.3%) than men (6.2%), and young adults aged 18 to 25 had the highest prevalence (18.6%) [59]. Resilience has been shown to be protective against daily and chronic stressors and to reduce the risk of depression—a risk factor for poor diet quality and related CVD outcomes [14,60,61,62,63,64,65,66,67,68,69]. Our findings and those of others suggest that resilience is significantly and positively associated with diet quality, even after adjusting for socioeconomic status and depression [46,70]. Thus, building on resilience, rather than focusing solely on deficit-based risk factors, may be a timely and important addition to dietary interventions, especially in younger adults [71].

In our study, early young adults (18–24 years of age) reported the lowest mean WELL Diet Scores, falling in the bottom half of our sample (58.7 ± 18.0 out of 120; IQR: 56–85). These had fewer resources, including significantly lower levels of education, were less likely to be married or to live with a partner, and reported slightly lower scores for measures related to mental health than older adults (including positive/negative emotions and overall well-being (see Table 1)). Several studies suggest that early young adults are targeted by and susceptible to marketing tactics for low-quality, affordable ultra-processed foods [72,73,74,75]. Perhaps younger adults are also less concerned about eating healthily due to their lower risk for diet-related diseases [76]. For example, just over 4% of early young adults reported hypertension in our sample. That percentage grew to 30% in late adults. Further research is warranted to determine whether resilience training, which includes stress management and resource mobilization, can lead to positive influences on the adherence to dietary guidelines for CVD prevention in young adults.

This study fills an important gap in the literature by providing valuable evidence on the relationship between diet quality and resilience throughout adulthood. However, the design, measures, and the generalizability of our findings have limitations. The cross-sectional design of our study inherently limits our ability to establish causal relationships between diet quality and resilience. This is because it does not allow us to assess changes over time, which are crucial for determining cause and effect. While we can examine associations, we are unable to determine whether resilience affects diet quality or vice versa. Our interpretation is that higher resilience leads to better diet quality, the primary outcome of our study.

Regarding our measures, SPRC experts developed several measures for the WELL study. In some cases, items from multiple measures were combined to assess one construct, or original scales were created. Although our adapted measures stem from validated scales, their modifications in our study context necessitate further validation to ensure their reliability and accuracy within our specific research framework. This is true for both our primary exposure (resilience) and main outcome (diet quality) measures. For instance, items from the Brief Resilience Scale and the Connor–Davidson Resilience Scale were combined to measure resilience, despite their differing definitions [77]. According to Ye and colleagues, the former is a belief-based construct, whereas the latter is a resource-based construct. Despite these differences, both scales have been associated with better diet quality in previous studies, are modifiable, and are therefore suitable for intervention [78,79,80]. In this way, their overlapping items could help to clarify the relationship between diet quality and resilience [29,35]. SPRC experts also developed the WELL Diet Score, which has been associated with the Alternative Healthy Eating Index 2010 (AHEI-2010), a diet quality measure proven to be a significant predictor of chronic disease risk [46,81,82].

Given the demographic composition of our study, which was primarily white and college-educated, the generalizability of our findings to the broader U.S. population might be somewhat limited. This is due to the potential underrepresentation of certain demographic groups in our sample. Despite the adaptation and merging of multiple measures, as well as a cumulative score for overall well-being, all WELL measures have been standardized so that they range from 0–10 to provide consistent estimates (described in Table 1). Compared with the national averages for the major racial/ethnic subgroups reported by the US Census 2020, our sample shows that Whites are correctly represented, Asians are overrepresented, and Blacks and Hispanics are underrepresented [83]. Compared with the national averages for the major racial/ethnic subgroups reported by the US Census 2020, our sample shows higher educational attainment [84]. With respect to mental health, we asked about depression over a lifetime compared with the two-week period in the SAMSHA assessment, which may have resulted in higher percentages. Despite this, our estimate may be higher than national averages. Unlike SAMHA, we also asked if depression was diagnosed by a health professional. Due to this, and the fact that depression is often undiagnosed, our estimates are likely to be conservative [85].

Furthermore, since the WELL study is focused on improving well-being, it may have attracted more health-conscious participants, yielding a selection bias. Even still, the diet quality scores remain suboptimal. This may highlight the need to conduct these studies in more racial and ethnic populations, especially those dealing with multiple forms of historical and contemporary oppression (e.g., racism, displacement, and socioeconomic disadvantage) that leave them particularly vulnerable to poor diet quality, stress, and other CVD-related risk factors [86,87].

Our study has numerous strengths and marks a significant contribution to the literature on diet quality and resilience. To the best of our knowledge, ours is the first to examine associations between diet quality and a formal measure of resilience across stages of adulthood in a US general population sample, representing both females and males who are in relatively good health. The primary outcome, of diet quality, is a comprehensive assessment designed to measure adherence to prevention guidelines, which are more robust than individual foods, food groups, or macronutrients. Our study also evaluated psychological resilience through two widely used measures that have been validated in US populations and that are associated with various physical and mental health outcomes, i.e., the Brief Resilience Scale and the Conner–Davidson Scale [88,89,90]. We had more sociodemographic diversity in our sample compared with previous studies, as well as resilience-related measures of well-being. As part of our analysis, we evaluated relevant sociodemographic and psychological confounders, such as education, depression, and positive affect. Another major strength of our study is the large sample size with a broad age range, which enables age stratification to explore potential age-related modifications in the relationship between diet quality and resilience.

## 5. Conclusions

We have provided foundational evidence of a significant and positive relationship between diet quality and resilience in adults across age groups. Our findings warrant further investigation into whether assessing and potentially intervening to improve resilience could help to increase the effectiveness of dietary interventions for CVD prevention, especially among young adults, who are the most vulnerable to poorer diet quality and stress management and, subsequently, to an earlier onset of other CVD risk factors. Notably, the relationship between resilience and diet quality is significant, even after adjusting for depression (despite its relatively high prevalence in our sample). This highlights the importance of addressing existing psychological assets in dietary intervention, as opposed to just deficit-based risk factors [91].

Accordingly, our findings have implications for strengths-based approaches to dietary interventions; namely, resilience increases with age, and it is modifiable and buildable. By raising awareness of existing levels of resilience before individuals engage in behavioral interventions and other psychological assets, we can create personalized behavioral plans. In turn, these can guide the practices of a wide range of healthcare professionals, including psychologists, dietitians, nurses, physicians, and behavior specialists, whose responsibilities include developing educational materials and care plans that encourage patients to adhere to dietary guidelines for chronic disease prevention and management. Also, behavioral research interventionists may benefit from our findings by incorporating resilience screens into behavioral health studies, particularly in populations with sociodemographic backgrounds that include exposure to multidimensional (multilevel, historical, and contemporary) stressors and trauma.

Considering our findings, resilience training may contribute to an increase in diet quality or an increased adherence to diet patterns for the prevention of cardiovascular disease (CVD), since it can buffer the negative effects of stress on diet quality. Understanding that additional research is needed in this area, as a next step, we will examine the relationship between diet quality and resilience resources in a diverse US representative sample of young African American women.

## Figures and Tables

**Figure 1 nutrients-16-01724-f001:**
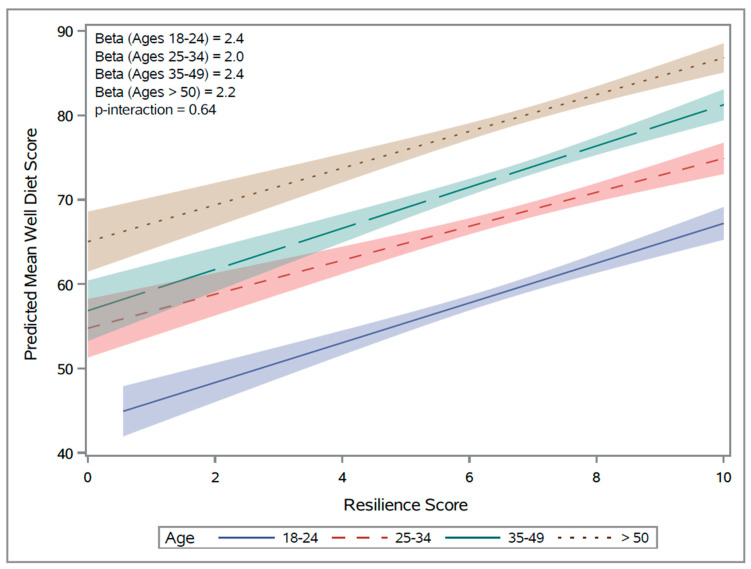
Resilience Score vs. predicted WELL Diet Score. Note that the unadjusted slopes are presented here based on the crude values.

**Table 1 nutrients-16-01724-t001:** Participant characteristics by stages of adulthood (quartile age groups in years).

Participant Characteristics		All Ages	Early Young	Late Young	Middle	Late
		(18–50+) N = 6171	(18–24) N = 1585	(25–34) N = 1485	(35–49) N = 1470	50+ N = 1631
Main outcome and primary exposure	WELL Diet Score, mean (SD) Range 0–120; higher score indicates higher diet quality	70.2 (19.3)	58.7 (18.0)	68.3 (17.8)	73.5 (17.6)	80.2 (17.2)
	Resilience, mean (SD) Range 0–10; higher score indicates more resilience	6.7 (1.7)	6.4 (1.7)	6.7 (1.8)	6.8 (1.7)	7 (1.7)
Sociodemographic	Age, mean (SD)	38.8 (16.8)	20.7 (2.1)	29.3 (2.8)	41.8 (4.7)	62.4 (8.4)
	*Gender, n (%) (n = 6148)*					
	Male	1668 (27.1)	554 (35.1)	365 (24.6)	311 (21.2)	438 (27)
	Female	4407 (71.7)	999 (63.2)	1094 (73.9)	1137 (77.7)	1177 (72.5)
	Transgender and genderqueer	73 (1.2)	27 (1.7)	22 (1.5)	16 (1.1)	8 (0.5)
	*Race/ethnicity, n (%) (n = 6120)*					
	White/Caucasian	3727 (60.9)	939 (59.5)	792 (53.6)	814 (55.8)	1182 (73.6)
	Hispanic	664 (10.8)	203 (12.9)	211 (14.3)	167 (11.4)	83 (5.2)
	Black/African American	268 (4.4)	103 (6.5)	49 (3.3)	60 (4.1)	56 (3.5)
	Asian	1523 (24.9)	369 (23.4)	449 (30.4)	424 (29.0)	281 (17.5)
	Other	165 (2.7)	52 (3.3)	45 (3.0)	39 (2.7)	29 (1.8)
	*Marital status, n (%) (n = 6146)*					
	Married	2558 (41.6)	60 (3.8)	496 (33.6)	954 (65.2)	1048 (64.6)
	Living with partner	564 (9.2)	122 (7.7)	263 (17.8)	106 (7.2)	73 (4.5)
	Single	2487 (40.5)	1388 (87.7)	677 (45.8)	254 (17.4)	168 (10.4)
	Other	537 (8.7)	13 (0.8)	42 (2.8)	149 (10.2)	333 (20.5)
	*Education, n (%) (n = 6135)*					
	HS or below	759 (12.4)	595 (37.6)	72 (4.9)	40 (2.7)	52 (3.2)
	Some college	1139 (18.6)	527 (33.3)	178 (12.0)	200 (13.7)	234 (14.5)
	College graduate	2030 (33.1)	381 (24.1)	654 (44.2)	459 (31.5)	536 (33.2)
	Graduate	2207 (36)	79 (5)	576 (38.9)	758 (52)	794 (49.1)
Perceived stress	Stress, mean (SD) (n = 6160) Range 0–10; higher score indicates greater stress	5.9 (1.7)	5.5 (1.7)	5.8 (1.6)	5.9 (1.7)	6.5 (1.7)
Lifestyle and BMI	Physical activity, mean (SD) (n = 6085) Range 0–10; higher score indicates more physical activity	5 (2.9)	4.8 (3.1)	5.1 (2.9)	4.9 (2.9)	5.3 (2.7)
	*Smoking status, n (%) (n = 6151)*					
	Never	5100 (82.9)	1445 (91.3)	1303 (88.0)	1169 (79.7)	1183 (72.9)
	Former	830 (13.5)	69 (4.4)	118 (8.0)	237 (16.2)	406 (25.0)
	Current	221 (3.6)	68 (4.3)	60 (4.1)	60 (4.1)	33 (2.0)
	BMI, mean (SD) (n = 6008)	25.2 (5.7)	24.1 (5.1)	24.9 (6.4)	25.8 (5.8)	25.9 (5.5)
	Self-reported history of hypertension, n (%) (n = 6025)	784 (13.0)	65 (4.3)	87 (6.0)	148 (10.2)	484 (30.0)
Mental health	Positive affect, mean (SD) (n = 6159) Range 0–10; higher score indicates more positive emotions	6.8 (1.7)	6.6 (1.7)	6.7 (1.6)	6.8 (1.7)	7.0 (1.7)
	Self-reported history of depression, n (%) (n = 5948)	1447 (24.3)	362 (24.5)	340 (23.8)	342 (23.6)	403 (25.3)
	Negative affect, mean (SD) (n = 6160) Range 0–10; higher score indicates fewer negative emotions (reversed-scored)	5.1 (1.8)	4.6 (1.8)	5.0 (1.7)	5.2 (1.8)	5.7 (1.7)
	Overall WELL Score (well-being), mean (SD) (n = 5918) Range 0–100; greater score indicates better overall well-being	66.2 (12.6)	62.6 (12.5)	65.2 (12.1)	66.8 (12.0)	70.2 (12.4)

**Table 2 nutrients-16-01724-t002:** Associations between resilience and the WELL Diet Score overall and by adulthood stage.

	Age-Adjusted	Early Young Ages 18–24	Late Young Ages 25–34	Middle Ages 35–49	Late Ages ≥ 50	*p*-Value for Interaction
Crude	2.3 ± 0.1 **	2.4 ± 0.3 **	2.0 ± 0.3 **	2.4 ± 0.3 **	2.2 ± 0.2 **	0.64
Model 1	2.1 ± 0.1 **	2.0 ± 0.2 **	1.9 ± 0.3 **	2.2 ± 0.3 **	1.9 ± 0.2 **	0.65
Model 2	1.5 ± 0.1 **	1.5 ± 0.3 **	1.1 ± 0.3 **	1.4 ± 0.3 **	1.5 ± 0.3 **	0.60
Model 3	1.3 ± 0.1 **	1.2 ± 0.3 **	1.3 ± 0.3 **	1.1 ± 0.3 **	1.3 ± 0.3 **	0.26
Model 4	1.2 ± 0.2 **	1.1 ± 0.3 **	1.2 ± 0.3 **	0.9 ± 0.3 *	1.0 ± 0.3 *	0.34
Beta ± Standard Error						

* *p* < 0.01 for all beta coefficients. ** *p* < 0.001 for all beta coefficients. Model 1 adjusts for sociodemographics (gender, race, education). Model 2 adjusts for sociodemographics and perceived stress. Model 3 adjusts for model 2 variables for sociodemographics, perceived stress, and lifestyle and BMI (physical activity, smoking status, and BMI). Model 4 adjusts for model 3 sociodemographics, perceived stress, lifestyle and BMI, and mental health (depression, positive affect).

## Data Availability

The original contributions presented in the study are included in the article/Appendix A, further inquiries can be directed to the corresponding author.

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
