# Peer review of "Diet Quality and Resilience through Adulthood: A Cross-Sectional Analysis of the WELL for Life Study"

_nutrients, 2024, doi:10.3390/nu16111724_

Round 1
Reviewer 1 Report
Comments and Suggestions for Authors
Dear Authors,
I hope this email finds you well.
I'm writing to you regarding your manuscript titled "Diet Quality and Resilience through Adulthood: A Cross-sectional Analysis of the WELL for Life Study," which you submitted to Nutrients for review. Your research delves into a fascinating and highly relevant topic, exploring the intricate interplay between dietary habits and psychological resilience across the various stages of adulthood. I've carefully reviewed your work, considering it from the perspective of a potential reader, and I believe it holds significant scientific value. However, I'd like to suggest some refinements that could enhance its clarity and impact.
1) One key aspect to emphasize more prominently is the inherent limitation of causal inference due to the cross-sectional nature of the study. This point should be made particularly clear in the Discussion section to ensure readers fully grasp the study's boundaries. Consider rephrasing the sentence as follows:
Original: "Due to the cross-sectional nature of this study, we are unable to determine causality as there is no temporal relationship."
Suggested: "The cross-sectional design of our study inherently limits our ability to establish causal relationships between diet quality and resilience. This is because it does not allow us to assess changes over time, which are crucial for determining cause and effect."
2) Upon examining the demographic characteristics of your sample (predominantly white and college-educated), I have some concerns about its representativeness of the broader American population. While I'm not an American myself, I've reviewed some statistics that suggest it would be beneficial to include a reference acknowledging this potential limitation in terms of external validity. Consider modifying the sentence as follows:
Original: "Our findings may not be generalizable to the non-WELL population."
Suggested: "Given the demographic composition of our study, primarily white and college-educated, the generalizability of our findings to the broader U.S. population might be somewhat limited. This is due to the potential underrepresentation of certain demographic groups in our sample."
3) The adaptation and combination of resilience scales require more in-depth validation. It would be valuable to highlight more directly that, while these measures are based on validated scales, they need further scrutiny within the specific context of your study. Consider rephrasing the sentence as follows:
Original: "Most of these adapted measures have not been validated, although they may have been constructed from validated measures."
Suggested: "Although our adapted measures stem from validated scales, their modifications in our study context necessitate further validation to ensure their reliability and accuracy within our specific research framework."
4) Lastly, I noticed the absence of DOIs in the references. These are essential according to the journal guidelines and provide direct access to the original studies. Please include them for completeness.
(Regarding self-citations, ensure they do not exceed 10% of the total references to maintain objectivity.)
I hope these suggestions are helpful.
Author Response
Comment to the Author:
1) One key aspect to emphasize more prominently is the inherent limitation of causal inference due to the cross-sectional nature of the study. This point should be made particularly clear in the Discussion section to ensure readers fully grasp the study's boundaries. Consider rephrasing the sentence as follows:
Original: "Due to the cross-sectional nature of this study, we are unable to determine causality as there is no temporal relationship."
Suggested: "The cross-sectional design of our study inherently limits our ability to establish causal relationships between diet quality and resilience. This is because it does not allow us to assess changes over time, which are crucial for determining cause and effect.
Authors’ Response:
Done.
Comment to the Author:
2) Upon examining the demographic characteristics of your sample (predominantly white and college-educated), I have some concerns about its representativeness of the broader American population. While I'm not an American myself, I've reviewed some statistics that suggest it would be beneficial to include a reference acknowledging this potential limitation in terms of external validity. Consider modifying the sentence as follows:
Original: "Our findings may not be generalizable to the non-WELL population."
Suggested: "Given the demographic composition of our study, primarily white and college-educated, the generalizability of our findings to the broader U.S. population might be somewhat limited. This is due to the potential underrepresentation of certain demographic groups in our sample."
Authors’ Response:
Done.
Comment to the Author:
3) The adaptation and combination of resilience scales require more in-depth validation. It would be valuable to highlight more directly that, while these measures are based on validated scales, they need further scrutiny within the specific context of your study. Consider rephrasing the sentence as follows:
Original: "Most of these adapted measures have not been validated, although they may have been constructed from validated measures."
Suggested: "Although our adapted measures stem from validated scales, their modifications in our study context necessitate further validation to ensure their reliability and accuracy within our specific research framework."
Authors’ Response:
Done.
Comment to the Author:
4) Lastly, I noticed the absence of DOIs in the references. These are essential according to the journal guidelines and provide direct access to the original studies. Please include them for completeness.
(Regarding self-citations, ensure they do not exceed 10% of the total references to maintain objectivity.)
Authors’ Response:
DOIs have been included to all references with a few exceptions because DOI were not available (e.g., census bureau reports).
Self-citations, do not exceed 10% of the total references.
Very helpful, thank you.

Reviewer 2 Report
Comments and Suggestions for Authors
Introduction:
Overall, very nicely written and referenced. One comment: in the first paragraph, you state 'it is less commonly recognized that diet tends to improve as we progress through phases of life' and then you follow with 'thus, improving diet quality is an important strategy for prevention and managing CVD etc'. I am not sure the point is clear here, why is improving diet quality as we age important if it already improves as we age?
Methods:
2.2.3, I think after your main groups, e.g. 'sociodemographic variables' or 'lifestyle and BMI variables' you should use a colon, rather than full stop.
It wasn't clear to me how the Total Well score was used from the methods. It is not mentioned as a covariate or in the statistical analysis. The description mentions 10 WELL domains, which is the first mention of these domains and it wasn't clear how these came about.
Results
I would not state 'in all ages, mean WELL Diet Scores were about 70 out of 120' and then put the exact amount in brackets - firstly, because this repeats what's in the table, and secondly, why say what it was roughly then follow up with what it was exactly. Instead, I would put something like 'WELL diet scores ranged from 58.7 in the youngest adults to 80.2 in late adulthood, with a mean for all ages of about 70/120.' I also suggest removing the statement 'substantial room for improvement' when talking about the WELL diet scores, and replacing with 'suggesting the overall diet quality could be improved in all age groups'. Then finally, you can remove the sentence immediately under the table, again repeating what's in the table.
I'm not sure why Figure 1 has the crude values - this probably needs to made clear at least in the footnotes, but I'd suggest showing model 4.
Discussion
I would make the point somewhere that the higher depression reported in your study may be closer to the actual rates, as it is likely there are many undiagnosed cases.
Well done, clearly presented and well reference throughout.
Author Response
Comment to the Author (on Introduction):
Overall, very nicely written and referenced. One comment: in the first paragraph, you state 'it is less commonly recognized that diet tends to improve as we progress through phases of life' and then you follow with 'thus, improving diet quality is an important strategy for prevention and managing CVD etc.' I am not sure the point is clear here, why is improving diet quality as we age important if it already improves as we age?
Authors’ Response:
The sentence now reads” “Thus, improving diet quality is a key protective strategy for preventing and managing CVD risk factors throughout life in early adults”
Comment to the Author (on Methods):
I think after your main groups, e.g. 'sociodemographic variables' or 'lifestyle and BMI variables' you should use a colon, rather than full stop.
It wasn't clear to me how the Total Well score was used from the methods. It is not mentioned as a covariate or in the statistical analysis. The description mentions 10 WELL domains, which is the first mention of these domains and it wasn't clear how these came about
Authors’ Response:
We have added colon after subtitles 'sociodemographic variables’ and 'lifestyle and BMI variables’.
The WELL Total Score was described under covariates mental health subsection of the Methods. See description below here.
For clarity, now the subtitle reads: Mental health (depression, positive and negative affect, and wellbeing):
It reads - The total WELL Score was, a measure of well-being, tabulated by combining scores for all 10 WELL domains, including social connectedness, lifestyle behaviors, physical health, stress and resilience, experience of emotions, purpose and meaning in life, sense of self, financial security, spirituality and religiosity, and exploration and creativity. Scores range from 0-100.
Given that the total WELL Score consists of several measures included in our primary analysis (like resilience and perceived stress) will did not include it in our statistical models.
Thus, the following line to was added to the statistical analysis section:
Note, Total WELL Score was not included in the statistical models.
Comment to the Author (on Results):
I would not state 'in all ages, mean WELL Diet Scores were about 70 out of 120' and then put the exact amount in brackets - firstly, because this repeats what's in the table, and secondly, why say what it was roughly then follow up with what it was exactly. Instead, I would put something like 'WELL diet scores ranged from 58.7 in the youngest adults to 80.2 in late adulthood, with a mean for all ages of about 70/120.'
I also suggest removing the statement 'substantial room for improvement' when talking about the WELL diet scores, and replacing with 'suggesting the overall diet quality could be improved in all age groups'.
Then finally, you can remove the sentence immediately under the table, again repeating what's in the table.
I'm not sure why Figure 1 has the crude values - this probably needs to made clear at least in the footnotes, but I'd suggest showing model 4.
Authors’ Response:
The sentence now reads:
Done.
The sentence now reads: WELL Diet Scores ranged from 58.7 in the youngest adults to 80.2 in late adulthood, with a mean for all ages of about 70 out of 120, suggesting the overall diet quality could be improved in all age groups.
The sentence immediately under the table 1 has been removed.
The following sentence has been added to the footnotes of Figure 1: Note that the unadjusted slopes are presented here based on the crude values.
Comment to the Author (on Discussion):
I would make the point somewhere that the higher depression reported in your study may be closer to the actual rates, as it is likely there are many undiagnosed cases.
Authors’ Response:
We have revised the following sentences to point out that depression is often undiagnosed and included corresponding citation.
“With respect to mental health, we asked about depression over a lifetime compared to the two-week period in the SAMSHA assessment, which may have resulted in higher percentages. Despite this, our estimate may be higher than national averages. Unlike SAMHA, we also asked if depression was diagnosed by a health professional. Due to this, and the fact that depression is often undiagnosed, our estimates are likely to be conservative.”
Handy, A., et al., Prevalence and Impact of Diagnosed and Undiagnosed Depression in the United States. Cureus, 2022. 14(8): p. e28011 DOI: 10.7759/cureus.28011.
Thank you!
